# OpenForecast v2: Development and Benchmarking of the First National-Scale Operational Runoff Forecasting System in Russia

**Georgy Ayzel** 

State Hydrological Institute, 199004 Saint Petersburg, Russia; ayzel@iwp.ru

**Abstract:** Operational national-scale hydrological forecasting systems are widely used in many countries for flood early warning systems and water management. However, this kind of system has never been implemented in Russia. OpenForecast v2—the first national-scale operational runoff forecasting system in Russia—has been developed and deployed to fill this gap. OpenForecast v2 delivers 7 day-ahead streamflow forecasts for 843 gauges across Russia. The verification study has been carried out using 244 gauges for which operational streamflow data were openly available and quality-controlled for the entire verification period (14 March–6 July 2020). The results showed that the developed system provides reliable and skillful runoff forecasts for up to one week. The benchmark testing against climatology and persistence forecasts showed that the system provides skillful predictions for most analyzed basins. OpenForecast v2 is in operational use and is openly available on the Internet.

**Keywords:** streamflow; runoff; forecasting; modeling; Russia

## 1. Introduction

Flood-related hazards are among the most devastating natural disasters [1]. The risks they pose are high for many economic sectors and tend to increase [1–3]. Furthermore, climate change amplifies the anticipated risks by making the hydrological cycle more intense and making the prediction of its processes more uncertain [4,5]. Thus, the mitigation of risks by developing timely and reliable streamflow forecasts remains a key focus of the research community [6,7].

Many state-of-the-art runoff forecasting systems operate at regional, national and global scales and provide forecasts at hourly, daily or (sub-) seasonal resolution [8–10]; for example, the European Flood Awareness System (EFAS [11]), the Global Flood Awareness System (GloFAS [12]), the FANFAR project (https://fanfar.eu/forecast/, last access: 19 November 2020), the forecasting system at the Australian Bureau of Meteorology [13,14], the Terrestrial Systems Modeling Platform (TSMP [15]), to name a few. These systems routinely derive skillful runoff forecasts that are used for—but are not limited to—early warning systems for extreme events and water management. The latest example of devastating floods in Mozambique in 2019 demonstrated the crucial role of timely and distributed forecasts to minimize human and economic losses [16].

Although global forecasting systems such as GloFAS may provide runoff forecasts for any river basin in the world, the efficacy of that forecast is limited. The reason for this is that the global systems are rarely calibrated against streamflow observations [17]. Moreover, the limited accessibility of regional hydrometric data could also hinder calibration [18]. Thus, the use of available regional information should be a key focus when developing new or advancing existing runoff forecasting services. That is particularly relevant for forecasting systems that operate at a local or national scale as they may explicitly consider the use of specific regional data by design [19–22].

Over recent decades, many countries or their territorial units (e.g., states or provinces) have developed their own runoff forecasting systems [8–10]. Corresponding services are available, e.g., in the US [23], UK [22,24], Australia [13,14], and New Zealand [25]. Still, many barriers hinder the development of runoff forecasting systems in many less developed countries worldwide, of which the most significant barriers are economic [26].

There is an increasing trend in the frequency of flood-related hazards in Russia [27]. However, no runoff forecasting systems operate at the national nor at the territorial level. This is surprising because the devastating flash-floods triggered by extreme rainfall (e.g., in Krymsk in 2012 [28]) and exceptional spring floods cause economic and human losses that are a few orders of magnitude higher than the investment required for establishing a national-scale forecasting system; it is all the more surprising as the building blocks comprising any modern runoff forecasting system—observational data, hydrological models, and numerical weather prediction (NWP) models—have been present in the public domain for years.

To fill this gap, in 2018, Ayzel et al. [21] developed the first version of OpenForecast—the first open-source operational runoff forecasting system in Russia (OpenForecast v1; https://hydrogo.github.io/openforecast/, last access: 20 November 2020). OpenForecast v1 has been in operation since 20 July 2018 and delivers a 3 day-ahead runoff forecast for two test-bed river basins: the Moskva River at Barsuki and Seraya River at Novinki. As an interim and forerunner system, OpenForecast v1 provided the guideline for the further development and scaling of the service [21]. In the presented paper, the development of the second version of OpenForecast (OpenForecast v2; https://openforecast.github.io/, last access: 20 November 2020) is discussed; it is the first national-scale operational forecasting system in Russia that delivers 7 day-ahead streamflow forecasts for 843 gauges across Russia. In particular, this study aims to present the development workflow for the established operational service and provide a benchmark for its efficiency in terms of two widely used conventional approaches: climatology and persistence [29].

Finally, the potential of openly available data and software that serve as a basis for forecasting system development is exploited. Similar to its predecessor, each component of OpenForecast v2 is freely and readily available, ensuring the replicability of the system as well as the reproducibility of the results.

## 2. Data

### 2.1. Streamflow and Water Level Observations

Archive streamflow (in $m^3/s$) and water level observations (in cm above the "gauge null") for hundreds of gauges across Russia are available at the website of the Automated Information System for State Monitoring of Water Bodies (AIS; https://gmvo.skniivh.ru, last access: 20 November 2020). The corresponding datasets are available for the period from 2008 to 2017 (10 years) and distributed in a machine-readable format (.csv).

Operational data are available only for water level observations at the Unified State System of Information website regarding the Situation in the World Ocean (ESIMO; http://esimo.ru/dataview/viewresource?resourceId=RU_RIHMI-WDC_1325_1, last access: 20 November 2020). The corresponding data are in open access but are available only for the last seven days. Thus, the author developed a script that routinely downloads all the available data and converts them to a machine-readable format (.csv). It should also be mentioned that operational data in ESIMO does not pass a quality control and may be inconsistent with data from the AIS system. The possible reasons for that inconsistency are the measurement instrument's change or the change of the "gauge null".

### 2.2. Meteorological Data

ERA5 global reanalysis [30] serves as a source of meteorological forcing variables: air temperature ($T$, °C) and precipitation ($P$, mm). ERA5 has a spatial resolution of $0.25° \times 0.25°$ and an hourly temporal resolution. ERA5 data are available for the period from 1979 to within five days of real-time and are taken from two sources: (1) quality-

controlled monthly updates, which are published within three months of real-time (namely ERA5), and (2) daily updates of the dataset, which are available within five days of real-time (namely ERA5T). All the data are available for download at the Climate Data Store (https://cds.climate.copernicus.eu, last access: 20 November 2020). Recently, ERA5 showed high reliability to be used as a reference dataset for hydrological modeling over North America [31].

ICON—the global NWP model of the German Weather Service (DWD) [32]—serves as a source of deterministic meteorological forecasts for air temperature and precipitation. ICON has a spatial resolution of about 13 km and a temporal resolution of 1 h and provides forecast data up to one week [32]. The data are freely available for download at the DWD's Open Data Portal (https://opendata.dwd.de/weather/nwp/icon/, last access: 20 November 2020).

In this study, ERA5 and ICON precipitation and air temperature data have been aggregated to the daily time step and then averaged at the basin scale for each available basin. The spatial averaging is based on the relative weights of the intersection between basin boundaries and corresponding grid cells. Potential evaporation ($PE$, mm) is calculated using the temperature-based equation proposed by Oudin et al. [33].

### 2.3. Gauge Attributes and Basin Boundaries

Gauge attributes, such as the identification number, name, location, and drainage area, are available in both AIS and ESIMO databases. However, the information for a large portion of gauges is inconsistent between these datasets. Thus, manual expert control has been done to assign a unique set of attributes to each presented gauge. Particular attention has been devoted to the correction of geographical position; i.e., gauge latitude and longitude. For example, some gauges did not match the river network, while some have a substantial difference between actual and reported (in AIS or ESIMO) drainage areas. All these inconsistencies have been manually corrected to derive a quality-controlled set of studied gauges.

For each gauge, the corresponding basin boundaries have been obtained using the standard GIS-instruments [34] and the MERIT Hydro digital elevation model [35]. There are 1004 gauges with basin areas from 10 to 100,000 km$^2$ in the compiled dataset's preliminary version.

## 3. Methods

### 3.1. OpenForecast Computational Workflow

The schematic illustration of the second version of the OpenForecast computational framework is shown in Figure 1. There is no conceptual difference between the current and preceding versions (for the latter, see Figure 4 in [21]). However, the specific differences are as follows:

1. Although the first version uses ERA-Interim reanalysis, the second one uses ERA5, a gradual development over ERA-Interim.
2. Although there is a single hydrological model—GR4J (in French, modèle du Génie Rural à 4 paramètres Journalier) [36]—that simulates runoff in the first version, the second version is complemented by the HBV (in Swedish, Hydrologiska Byråns Vattenbalansavdelning) hydrological model [37,38].
3. Although the single loss function of the Nash–Sutcliffe efficiency coefficient (NSE [39]) has been used for model calibration in the first version, the second version is complemented using the Kling–Gupta efficiency coefficient (KGE [40]).
4. While the first version of the system derives a forecast for three days ahead, the second version extends this to seven days.
5. The number of gauges increased from two in the first version to 843 in the second version.

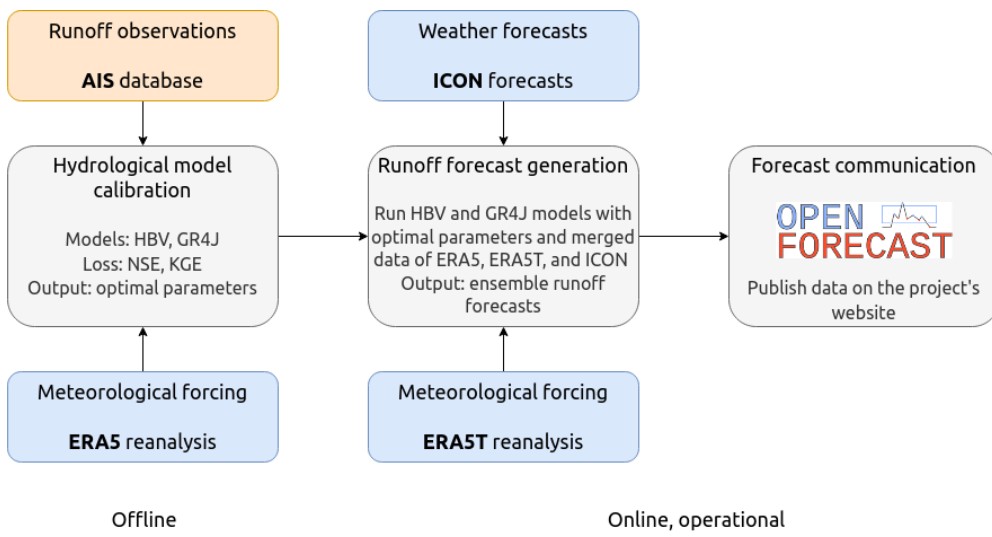

**Figure 1.** Illustration of the OpenForecast computational workflow.

The computational workflow is introduced in more detail in the following subsections, describing the underlying computational procedures in a step-by-step fashion. Figure 1 shows that the computational procedures comprising the OpenForecast's workflow can be classified into two types: (1) offline, which are done once and require only historical data, and (2) online, which run on an everyday basis and require new operational data each time.

### 3.1.1. Model Calibration

The core offline procedure is a hydrological model calibration. In the presented study, I use two conceptual lumped hydrological models: HBV [37,38] and GR4J [36]. Although the HBV model has an internal module for representing snow-related processes, the GR4J model has been coupled with the Cema–Neige snow accumulation routine [41,42]. Both models represent the water balance at the basin scale using storage reservoirs (three in HBV and two in GR4H) and require only daily precipitation, air temperature, and potential evaporation as inputs (see Section 2.2). HBV and GR4J models have 14 and six free parameters, respectively (Tables 1 and 2).

**Table 1.** Description and calibration ranges for GR4J model parameters (based on Ayzel et al. [21]).

| Parameters | Description | Calibration Range |
|:---:|:---:|:---:|
| X1 | Production store capacity (mm) | 0–3000 |
| X2 | Intercatchment exchange coefficient (mm/day) | −10–10 |
| X3 | Routing store capacity (mm) | 0–1000 |
| X4 | Time constant of unit hydrograph (day) | 0–20 |
| X5 | Dimensionless weighting coefficient of the snowpack thermal state | 0–1 |
| X6 | Day-degree rate of melting (mm/(day*°C)) | 0–10 |

For each basin, the optimal values of model parameters have been found by their calibration (numerical optimization) against the entire period of observed runoff time series [43] (see Section 2.1). To this end, a global optimization algorithm of differential evolution is used [44]. This algorithm finds a set of optimal model parameters by minimizing the loss function, which is either $1 - NSE$ or $1 - KGE$, where NSE and KGE are the Nash–Sutcliffe [39] and Kling–Gupta efficiency coefficients [40], respectively. The calibration ranges of model parameters (Tables 1 and 2) have been adopted from several studies [42,45–49]. The calibration procedure for each basin ends up with four optimal sets of model parameters, resulting from the use of two models (HBV, GR4J)

and two loss functions (NSE, KGE). The model codes and corresponding optimization routines are openly available as a part of the Lumped Hydrological Model Playground (https://github.com/hydrogo/LHMP, last access: 23 November 2020).

**Table 2.** Description and calibration ranges for HBV model parameters (based on Beck et al. [45]).

| Parameters | Description | Calibration Range |
| --- | --- | --- |
| TT | Threshold temperature when precipitation is simulated as snowfall (°C) | −2.5–2.5 |
| SFCF | Snowfall gauge undercatch correction factor | 1–1.5 |
| CWH | Water holding capacity of snow | 0–0.2 |
| CFMAX | Melt rate of the snowpack (mm/(day*°C)) | 0.5–5 |
| CFR | Refreezing coefficient | 0–0.1 |
| FC | Maximum water storage in the unsaturated-zone store (mm) | 50–700 |
| LP | Soil moisture value above which actual evaporation reaches potential evaporation | 0.3–1 |
| BETA | Shape coefficient of recharge function | 1–6 |
| UZL | Threshold parameter for extra outflow from upper zone (mm) | 0–100 |
| PERC | Maximum percolation to lower zone (mm/day) | 0–6 |
| K0 | Additional recession coefficient of upper groundwater store (1/day) | 0.05–0.99 |
| K1 | Recession coefficient of upper groundwater store (1/day) | 0.01–0.8 |
| K2 | Recession coefficient of lower groundwater store (1/day) | 0.001–0.15 |
| MAXBAS | Length of equilateral triangular weighting function (day) | 1–3 |

### 3.1.2. Generation of Runoff Forecast

The obtained sets of optimal model parameters form a core for further computational procedures of runoff forecast generation. At this stage, a hydrological model was run with a set of optimal parameters, providing coupled archive (ERA5) and operational (ICON) meteorological time series as input (Figure 2). As the ERA5T data has a latency of 5 days (see Section 2.2), the seamless integration of (archive) ERA5 and (operational) ICON data at forecast time *t* cannot be ensured: there will always be a gap of at least five days. In this way, the use of ICON hindcasts is proposed, which are forecast from previous days to fill the corresponding gap. However, to back up the presented workflow from unpredictable delivery delays of ERA5T data, the period filled with ICON hindcasts is extended to 7 days (t−7 days in Figure 2). Thus, for each river basin (or gauge), the output of the runoff forecast generation stage is an ensemble of four runoff forecasts, two of which are simulated with the HBV model driven with two sets of model parameters (calibrated against NSE and KGE, respectively), and the other two are simulated with the GR4J model and the corresponding sets of its optimal parameters.

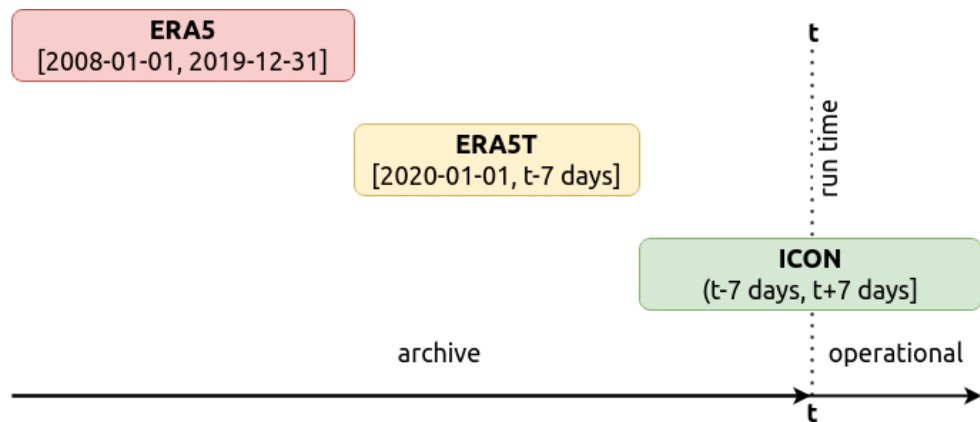

**Figure 2.** Illustration of data flows during the process of runoff forecast generation.

### 3.1.3. Forecast Communication

The simulated four-realization ensemble of runoff forecasts must then be properly communicated to a broad audience. To this end, for each river basin, the bokeh Python library (https://docs.bokeh.org, last access: 23 November 2020) was used to generate a webpage that consists of an interactive plot showing 7 day-ahead runoff forecast, as well as the hindcast for the proceeding seven days (Figure 3). Individual realizations of the forecast are not presented; instead, the ensemble mean (bold line in Figure 3) and ensemble spread (filled area in Figure 3) are displayed. There are numerous approaches to runoff forecast visualization [50]. However, this setup proved its reliability in the first version of OpenForecast [21]. The main webpage of OpenForecast v2 demonstrates all the gauges for which runoff forecasts are available on an interactive map (https://openforecast.github.io/, last access: 29 November 2020).

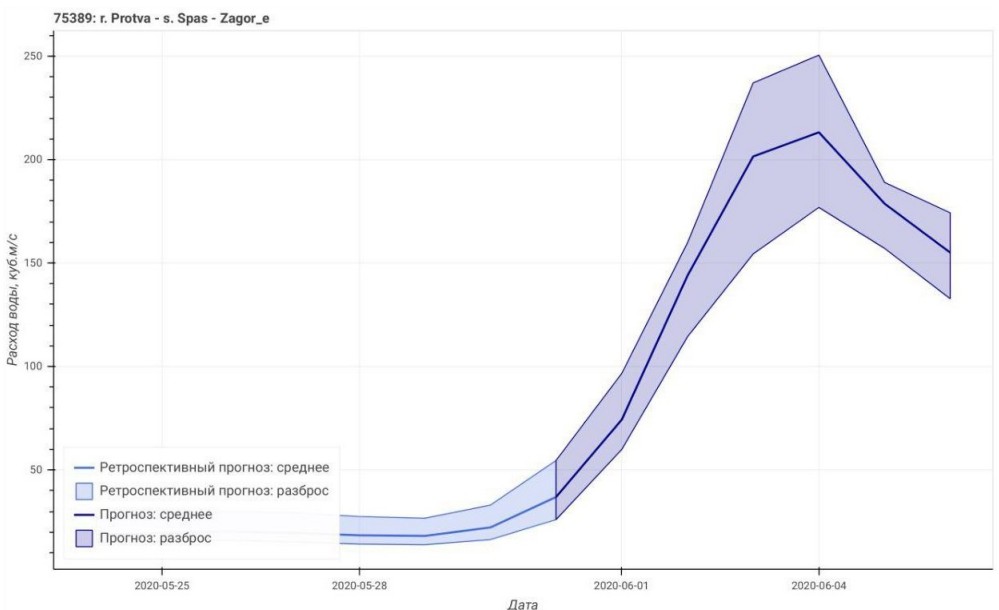

**Figure 3.** Example of the issued forecast for the Protva River at Spas-Zagor'e. Transcription from Russian as follows. *x*-axis: date; *y*-axis: discharge, $m^3/s$. The legend shows the following: bold light blue line: hindcast mean; light blue area: hindcast spread; bold blue line: forecast mean; blue area: forecast spread.

### 3.1.4. Computational Details

OpenForecast v2 has been in operational use since 14 March 2020. The OpenForecast computational workflow (Figures 1 and 2) runs on a webserver with a single CPU (two computational threads) and 4 Gb of memory. The forecast cycle begins daily at 07:00 UTC to produce runoff forecasts for the set of used gauges for the next 7 days (the day of the initial run is included). All the computations take around 20 min. Similar to the first version, the second version of the OpenForecast computational framework entirely relies on open-source software packages. The Python programming language (https://www.python.org/, last access: 23 November 2020) is used in this work with several open-source libraries: cdsapi (https://github.com/ecmwf/cdsapi, last access: 23 November 2020), xarray (http://xarray.pydata.org, last access: 23 November 2020), cfgrib (https://github.com/ecmwf/cfgrib, last access: 23 November 2020), and cdo (https://code.mpimet.mpg.de/projects/cdo/, last access: 23 November 2020) to preprocess ERA5 and ICON data, numpy (https://numpy.org/, last access: 23 November 2020), pandas (https://pandas.pydata.org/, last access: 23 November 2020), and geopandas (https://geopandas.org/, last access: 23 November 2020) for general-purpose data analysis and calculation, scipy (https://www.scipy.org/, last access: 23 November 2020) for implementation of global optimization algorithm of differential evolution, and matplotlib (https:

//matplotlib.org/, last access: 23 November 2020), bokeh (https://docs.bokeh.org, last access: 23 November 2020), and folium (https://python-visualization.github.io/folium/, last access: 23 November 2020) for plotting.

### 3.2. Benchmarks and Verification Setup

The forecasts from operational systems are typically evaluated in terms of the degree of their similarity with observations [29,51]. To this end, here, three efficiency metrics that are widely used in hydrological studies are employed: the Nash–Sutcliffe Efficiency coefficient (NSE; Equation (1); [39]), the Kling–Gupta Efficiency coefficient (KGE; Equation (2); [40]), and the systematic error (BIAS, Equation (3)).

$$NSE = 1 - \frac{\sum_\Omega (Q_{sim} - Q_{obs})^2}{\sum_\Omega (Q_{obs} - \overline{Q_{obs}})^2} \tag{1}$$

$$KGE = 1 - \sqrt{(r-1)^2 + (\frac{\sigma_{sim}}{\sigma_{obs}} - 1)^2 + (\frac{\overline{Q_{sim}}}{\overline{Q_{obs}}} - 1)^2} \tag{2}$$

$$BIAS = \frac{\sum_\Omega (Q_{sim} - Q_{obs})}{\sum_\Omega (Q_{obs})} \times 100, \% \tag{3}$$

where $\Omega$ is the period of evaluation, $Q_{sim}$ and $Q_{obs}$ are the simulated and observed runoff, $\overline{Q_{sim}}$ and $\overline{Q_{obs}}$ are the mean simulated and observed runoff, $r$ is the correlation component represented by Pearson's correlation coefficient, $\sigma_{sim}$ and $\sigma_{obs}$ are the standard deviations in simulations and observations, and $\text{cov}_\Omega (Q_{obs}, Q_{sim})$ is the covariance of simulated and observed runoff. NSE and KGE are positively oriented and not limited at the bottom: a value of 1 represents a perfect correspondence between simulations and observations. According to Knoben et al. [52], NSE > 0 and KGE > −0.41 can be considered to be showing efficacy against the mean flow benchmark. Bias is unbounded with the perfect value of 0.

Another important component of the forecast evaluation is whether the forecasts add value or show skill compared to some benchmark [29,53]. Thus, forecast skill (Equation (4)) can be assessed by the direct comparison of the forecast and the benchmark accuracy for a given efficiency metric.

$$Skill = \frac{A_{forecast} - A_{benchmark}}{A_{perfect} - A_{benchmark}} \tag{4}$$

where $A_{forecast}$ is the forecast accuracy in terms of some efficiency metric, $A_{benchmark}$ is the benchmark accuracy in terms of the same efficiency metric, and $A_{perfect}$ is the perfect accuracy for this metric; i.e., 1 for NSE and KGE, and 0 for bias. Skill score values are assigned qualitative descriptions, such as having positive (skill > 0) and negative (skill < 0) skill [14,54].

Following Pappenberger et al. [29], two benchmarks are considered here: (runoff) persistence and climatology. The rationale behind the use of the corresponding benchmarks and computational details are as follows.

- Runoff climatology(hereafter climatology) is a naive benchmark that requires only information about historical runoff observations. From the general public's perspective, this benchmark can be formulated as "The situation will be the same as in the year *YYYY*". Although the climatology benchmark can be dynamically calculated for each date of the forecast, here, the use of an a posteriori estimate is proposed; i.e., the single-year realization from the available 10-year climatological sample (2008–2017) that has the highest correlation coefficient with observations from the verification period. In this way, the climatology benchmark here will be "the best guess" one can make based on the available climatological sample; i.e., without any forecasting system at all.

- The runoff persistence (hereafter persistence) benchmark belongs to the change-signal category of benchmarks. It assumes that for any lead time, the runoff will be the same as the last observation (at forecast time). Despite its simplicity, persistence may be useful for short-range forecasting where the forecast signal is dominated by the auto-regression of flow [29]. Following Pappenberger et al. [29], "the last observation" is not considered here as a measured discharge, rather as the last runoff prediction simulated by the hydrological model. That choice ensures consistency and offers a homogeneous verification data set that is usually not readily available for operational observations. Thus, persistence shows the gain provided by the use of a deterministic meteorological forecast.

The verification period is considered from 14 March 2020 (the launch of OpenForecast v2) to 6 July 2020. For this period, water level observations were collected using the ESIMO system (Section 2.1). Then, for each gauge, the water level was transformed to discharge based on the stage–discharge relationship, which was calculated based on available historical observations from the AIS system (Figure 4). To find a rating curve, which is a functional relationship between water level ($H$) and discharge ($Q$), three polynomial approximations—linear ($Q = aH + b$), quadratic ($Q = aH^2 + bH + c$), and cubic ($Q = aH^3 + bH^2 + cH + d$)—have been fitted to observations (Figure 4). The approximation with the lowest mean absolute error (MAE) has been selected for each gauge. To minimize the effect of the temporal transformation of the rating curve [55], only the last 365 pairs of water level and discharge observations were considered to rate the curve calculation.

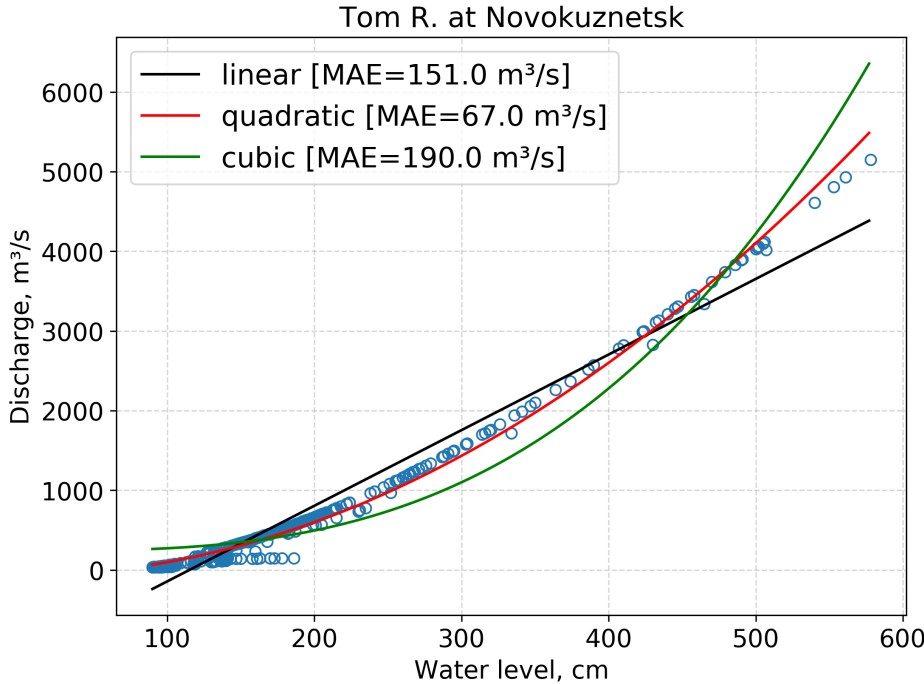

**Figure 4.** Example of rating curve calculation (Tom River at Novokuznetsk).

## 4. Results and Discussion

### 4.1. Hydrological Model Calibration

The calibration procedure (Section 3) was carried out for each basin from the set of 1004 available river basins (Section 2.3). In detail, two hydrological models, HBV and GR4J, were calibrated using two loss functions, KGE and NSE, for automatic fitting. That resulted in four configurations of "model–optimal parameter" pairs: $HBV_{NSE}$, $HBV_{KGE}$, $GR4J_{NSE}$, and $GR4J_{KGE}$. For example, $HBV_{NSE}$ represents the HBV model with optimal parameters derived from the calibration using NSE as a loss function. Following recommendations in

Knoben et al. [52], only river basins for which each model configuration passed both thresholds for "skillful" predictions—i.e., NSE > 0 and KGE > −0.41—were retained. This choice led to the selection of 843 river basins (gauges) that form the core of OpenForecast v2. The database that contains all the optimal model parameters for these basins is readily available in an open repository (https://doi.org/10.5281/zenodo.4328996, last access: 17 December 2020) and could be used for further analysis beyond the scope of the presented study.

Figure 5 demonstrates the calibration results for the core set of 843 basins in terms of NSE and KGE metrics. The median efficiencies in terms of NSE and KGE metrics are 0.78 and 0.84; 0.76 and 0.87; 0.77 and 0.81; 0.74 and 0.86 for $HBV_{NSE}$, $HBV_{KGE}$, $GR4J_{NSE}$, and $GR4J_{KGE}$, respectively. According to Moriasi et al. [56] and Knoben et al. [52], runoff simulations can be considered to be satisfactory if NSE > 0.5 or KGE > 0.3, respectively. Based on that, at least 725 (86%; $GR4J_{KGE}$ model configuration) and 841 (99.8%; $HBV_{NSE}$ model configuration) basins passed the corresponding thresholds for NSE and KGE, respectively. Thus, the obtained calibration results are promising: even with meteorological reanalysis data as forcing elements, both hydrological models showed a reliable efficiency for runoff simulation at a daily temporal resolution. That confirms recent findings [18,31,46–48,57,58], which also showed the high potential of modern climate reanalysis data for use in hydrological modeling applications. Furthermore, the calibration results confirmed that the used model configurations met a common prerequisite that ensures a strong ensemble—efficient yet different model configurations [59].

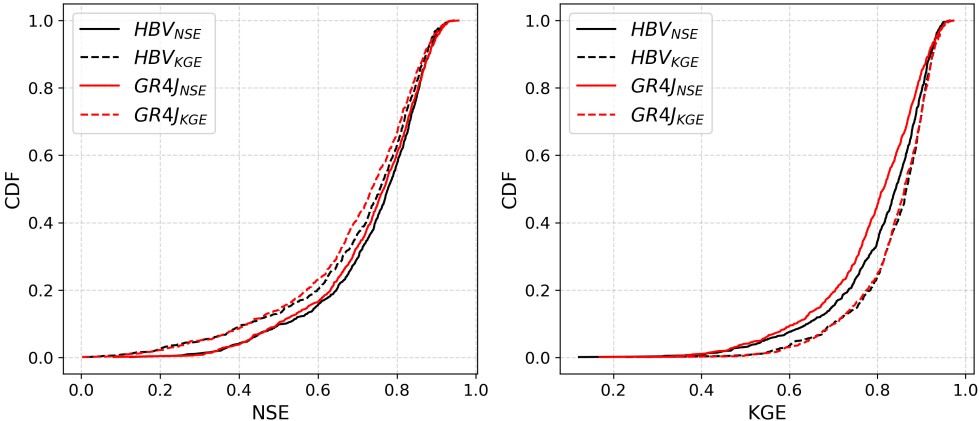

**Figure 5.** Cumulative density functions (CDF) for the NSE (**left plot**) and KGE (**right plot**) for all model configurations.

### 4.2. Selection of Reference Gauges

For the verification period (14 March 2020 to 6 July 2020; 115 days), water level observations were collected for every gauge present in the ESIMO system (Section 3.2). Then, a detailed analysis of collected data was performed to select the set of reference gauges, where observed water levels were reliable and consistent with those from the AIS database (Section 2.1). The corresponding analysis included automated tests (e.g., detection of outliers, sudden changes in flow dynamics, correspondence with historical observations), and the manual visual inspection of data. This detailed yet subjective analysis led to 244 gauges being selected from the ESIMO database, which were suitable for OpenForecast v2 verification (Figure 6). The following verification results are shown only for the specified set of gauges (river basins). The selection made excluded many gauges from the further analysis, but I argue that the remaining 244 gauges, which represent 29% out of 843 gauges in operation in OpenForecast v2, form a factual basis for a comprehensive analysis and benchmark of the OpenForecast streamflow forecasting system.

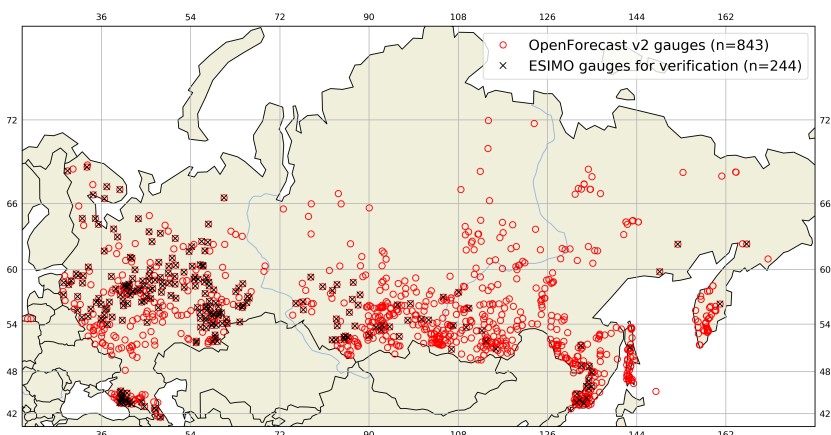

**Figure 6.** The spatial location of OpenForecast v2 gauges (n = 843) and those from the ESIMO database that were selected for the verification procedure (n = 244).

### 4.3. Benchmark and Verification Results

The set of evaluation metrics (Section 3.2) was calculated for each model configuration and for each gauge in the reference set (Figure 6, Section 4.2). Although individual forecasts produced by different model configurations—i.e., $HBV_{NSE}$, $HBV_{KGE}$, $GR4J_{NSE}$, and $GR4J_{KGE}$—are available for analysis, in the following subsection, only the analysis of the ensemble means of individual forecasts is presented. The ensemble mean (hereafter ENS) is calculated as an arithmetic mean over individual forecast members and represents the average agreement over them. Furthermore, it is the ensemble mean that is communicated as "runoff prediction" (both forecast and hindcast) in the OpenForecast system (Figure 3, Section 3.1). Thus, this choice is also justified by user experience, as ENS represents the actual forecast, rather individual members forming its uncertainty interval.

Figure 7 demonstrates the ensemble mean forecast (ENS) performance in terms of NSE and KGE metrics as a function of lead time for the set of reference basins. The visually distinct results show that (1) ENS shows a reliable median efficiency over the lead time of 7 days but (2) a gradual loss of efficiency over lead time is present, as expected; (3) the loss of efficiency is more pronounced for the NSE metric, both for a median value and width of the Interquartile Range (IQR). This loss of efficiency is usually referred to as the deterioration of the efficiency of meteorological forecast [8,10,21,29]. However, other factors, such as the cumulative effect of biased initial conditions, may also play a considerable role in the corresponding efficiency loss [29,60]. Still, both median NSE and KGE demonstrate values that are higher than behavioral values: 0.5 for NSE and 0.3 for KGE (after Knoben et al. [52]).

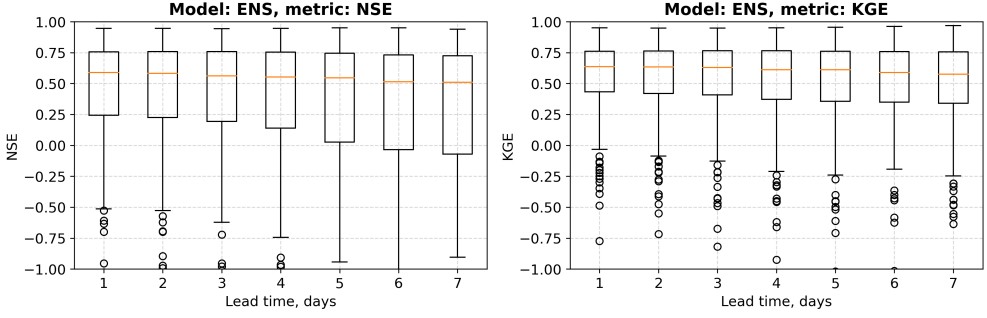

**Figure 7.** Verification of the ensemble mean forecast (ENS) in terms of NSE (**left plot**) and KGE (**right plot**) for the entire verification period and the entire set of reference gauges. The boxplot box represents the interquartile range (IQR, the difference between the 25th and 75th quantiles); the whiskers represent ±1.5×IQR from the 25th and 75th quantiles, respectively; the yellow line denotes the median value.

To distinguish the difference in ENS forecast efficiency between river basins of different sizes, all the reference basins were classified into three groups based on basin area: small (<1000 km$^2$), medium (1000–10,000 km$^2$), and large (>10,000 km$^2$). Figure 8 demonstrates the efficiency (in terms of NSE and KGE metrics) of ENS forecast as a function of lead time for reference basins of different sizes. The loss of the forecast efficiency over lead time is more pronounced for small rather than for medium or large basins. Thus, the loss of efficiency from the first to a seventh day-ahead forecasts (in terms of median NSE) for small, medium, and large basins are 0.68, 0.06, and 0.04, respectively. For KGE, these differences are less pronounced, yet still present. Thus, forecasts issued for small basins are less reliable than those for medium and large basins. This is expected as smaller basins tend to react to the changes in weather or/and initial conditions more rapidly [29,61]. Thus, advancing runoff predictions at small spatial scales must have a higher priority for the hydrological modeling community [62].

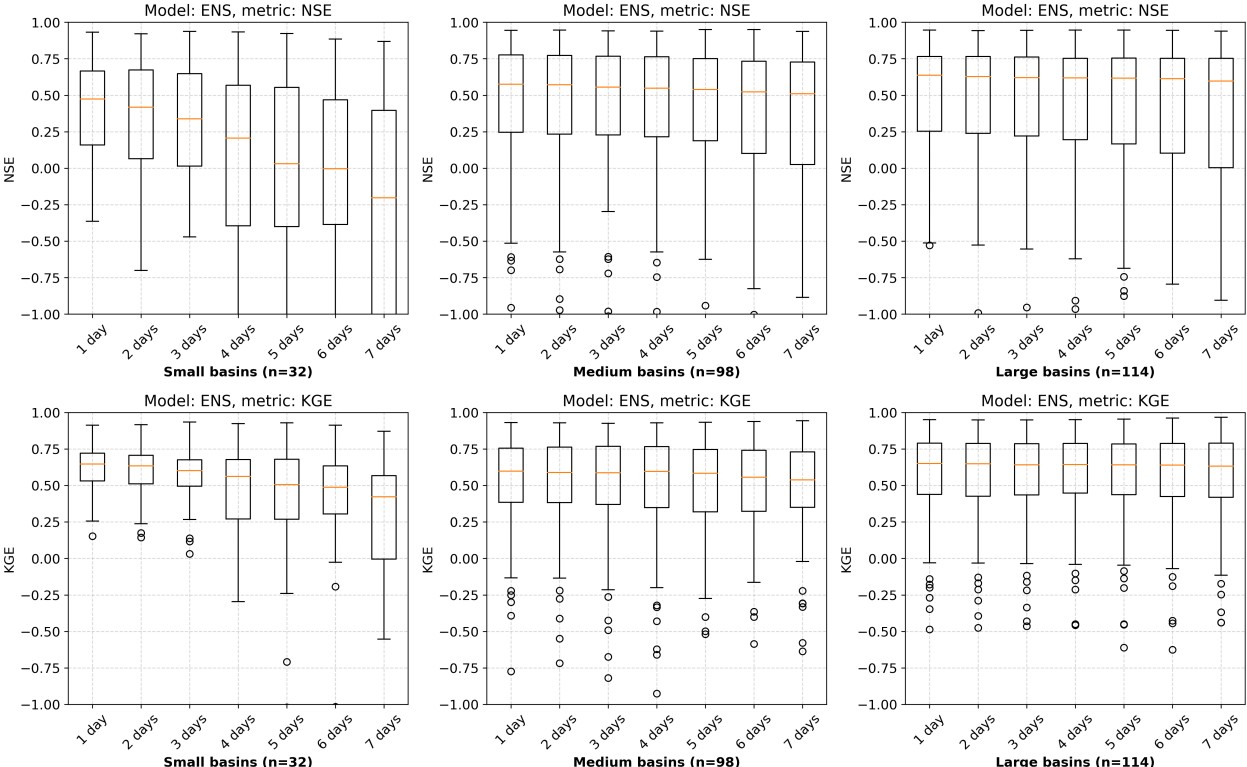

**Figure 8.** Verification of the ensemble mean forecast (ENS) in terms of NSE (top row) and KGE (bottom row) for the three groups of river basins: small (left column), medium (middle column), and large (right column). The boxplot box represents the Interquartile Range (IQR, the difference between the 25th and 75th quantiles); the whiskers represent ±1.5×IQR from the 25th and 75th quantiles, respectively; the yellow line denotes the median value.

Although the results of OpenForecast v2 efficiency (Figures 7 and 8) are promising, it is important to estimate the gain provided by OpenForecast v2 compared to two standard benchmarks (Section 3.2): climatology and persistence. To this end, efficiency metrics were compared for both ensemble mean forecast (ENS) and the set of benchmark models; then, respective skill scores were calculated for each basin (Equation (4)). Figure 9 shows whether OpenForecast forecasts were skillful in comparison to climatology (skill > 0) for the set of reference gauges or not (skill < 0). For the sake of brevity, only the skill for lead times of one, three, and seven days is plotted. Results show that for the lead time of one day, the provided forecasts have a positive skill for at least 86% (i.e., for 204 out of 244 basins in terms of NSE) of basins from the reference set. There is a slight decrease in the number of skillful basins with lead time. For the lead time of 7 days, forecasts are skillful for at least 78% of analyzed basins (i.e., for 191 out of 244 basins in terms of KGE). The results

do not clearly represent the expected pattern of less skillful predictions in highly seasonal basins [29], where climatology may provide a solid benchmark estimate [63]: there are many adjacent basins across the entire spatial coverage of data where the calculated skills have opposite signs. Moreover, there are also many basins in which the calculated skill may have different signs for calculated metrics (e.g., positive skill in terms of NSE and negative in terms of bias). Thus, the estimate of the number of basins where climatology is skillful compared to OpenForecast is optimistic. Furthermore, it should be mentioned that the implementation of the climatology benchmark in the presented study tends to provide a more optimistic assessment of its efficiency because of a posteriori calculation (Section 3.2).

The climatology benchmark remains efficient for a considerable portion of basins. However, the overall positive skill provided by OpenForecast is clear. Moreover, the skill of the climatology benchmark tends to decrease due to climate change. According to recent studies, the number of extreme events will increase in a warmer climate [64–66]; thus, runoff climatology will give less efficient estimates in the future. The rationale behind the use of the climatology benchmark was from the comparison of two scenarios—(1) when a runoff forecasting system was present, and (2) when there was no runoff forecasting system—and one could make predictions based only on an analogy, the naive version of which is "The situation will be the same as in the year *YYYY*". The obtained results confirmed that the forecasting system (OpenForecast) ensures the production of more reliable runoff forecasts in comparison to naive approaches. Thus, the urgent need for the development and further distribution of such systems for operational runoff forecasting is confirmed.

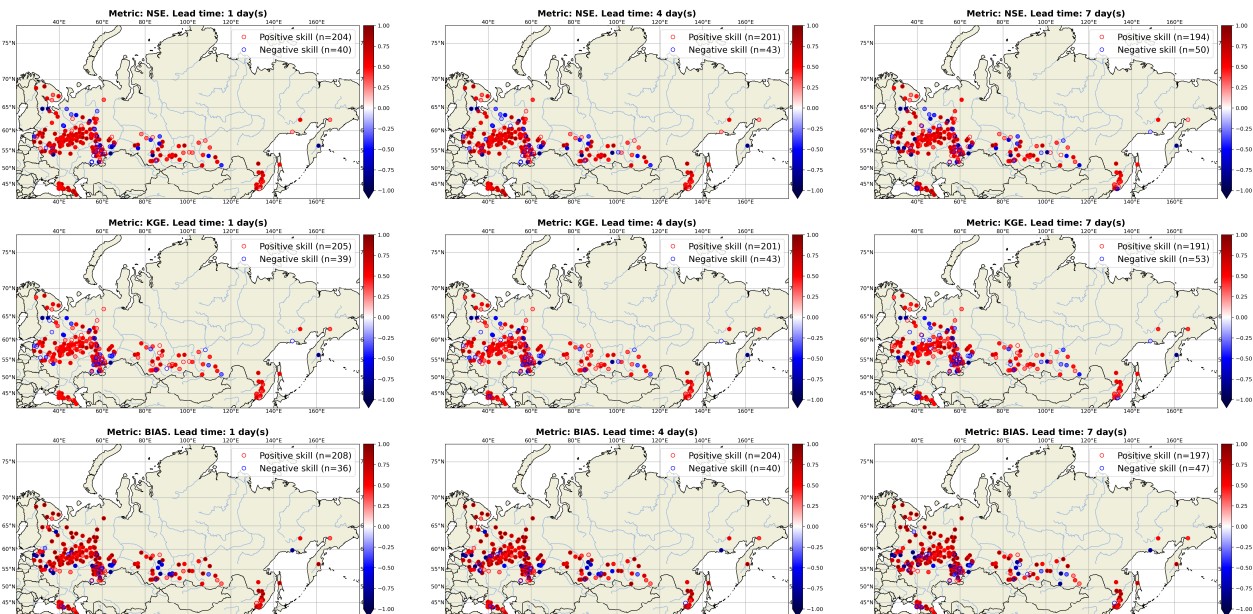

**Figure 9.** The skill of OpenForecast v2 with NSE (top row), KGE (middle row), and bias (bottom row) metrics for forecasts against the climatology benchmark for lead times of one (left column), three (middle column), and seven (right column) days.

Similar to Figure 9, Figure 10 demonstrates the skill of the ensemble mean (ENS) forecast over the runoff persistence benchmark. As expected, runoff forecasts are less skillful compared to persistence than climatology. However, even for the lead time of one day, at least 62%, 57%, and 53% of basins from the reference set demonstrate positive skill in terms of NSE, KGE, and bias, respectively. The percentage of skillful basins progressively increases with lead time. For the lead time of seven days, it reaches 80%, 89%, and 66% in terms of NSE, KGE, and bias, respectively. Moreover, the skill also increases in absolute values, becoming more pronounced with increasing lead time.

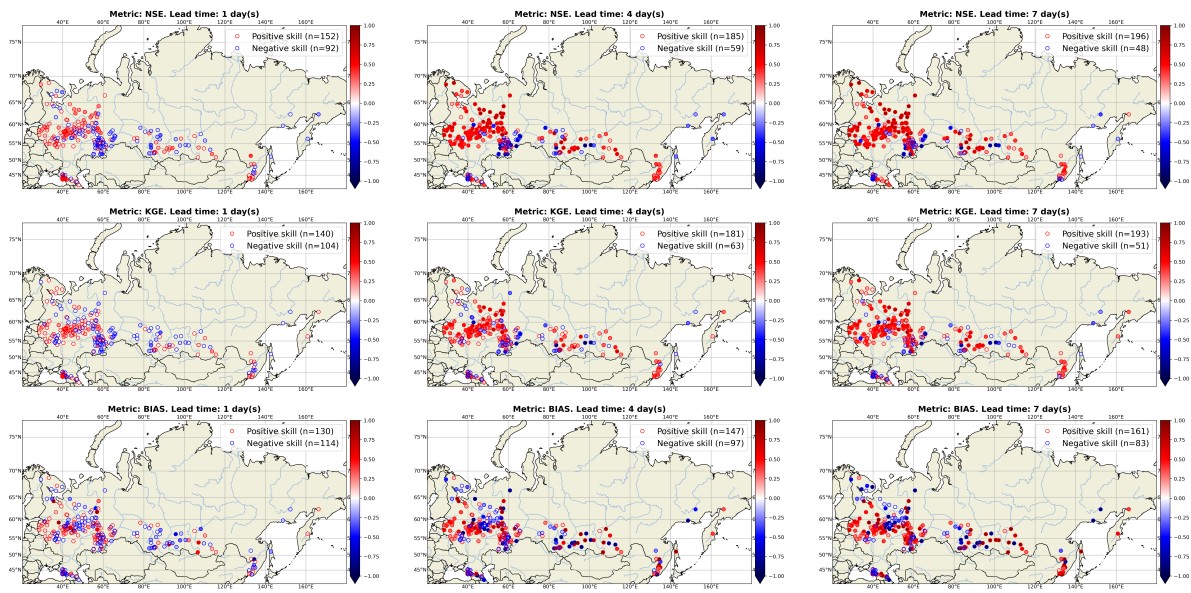

**Figure 10.** The skill of OpenForecast v2 with NSE (top row), KGE (middle row), and bias (bottom row) metrics for forecasts compared to the persistence benchmark for lead times of one (left column), three (middle column), and seven (right column) days.

The obtained results indicate the crucial value of NWP weather forecast use in the OpenForecast system. Although persistence is a skillful benchmark for very short lead times, it becomes less reliable for longer lead times [7,29]. Thus, the forecast skill compared to persistence benefits more from the use of NWP weather forecasts with increasing lead time. In this way, the further development of runoff forecasting systems should focus on the reliability of weather forecasts due to their high effect on the resulting efficiency.

In summary, the verification results showed the high reliability, skill, and additional value provided by the OpenForecast v2 system. The current computational workflow (Section 3.1, Figure 1) proved its efficiency and demonstrated remarkable progress in comparison to the first version of the OpenForecast system [21]. For example, the verification results of the OpenForecast v1 system showed its limited ability to simulate flood volumes. To compensate for such errors, a data assimilation routine has been developed to update forecasts based on operational streamflow observations; that led to a significant increase in forecast efficiency, from negative values to 0.9 in terms of NSE. Instead, the second version of the OpenForecast system does not include any data assimilation routine yet provides reliable predictions (Figures 7 and 8). I argue that two components ensure the demonstrated high out-of-the-shelf efficiency of OpenForecast v2:

1. The transition from ERA-Interim to ERA5 meteorological reanalysis (including the use of ERA5T product; Section 2.2).
2. The transition from deterministic to ensemble runoff forecast, which is produced by different hydrological model configurations (Section 3.1).

The verification of OpenForecast v2 proved the reliability of the framework on which it is based. However, there are several issues worth further discussion. In detail, I acknowledge that OpenForecast efficiency can be limited by the following factors:

1. The use of meteorological reanalysis data instead of observation-based products.
2. The use of non-homogeneous meteorological data—i.e., ERA5 reanalysis—but ICON NWP forecast.
3. The lack of observational streamflow data assimilation.
4. The lack of an error correction routine.
5. The use of lumped conceptual hydrological models, while far more advanced models exist.

The aforementioned factors could have a major impact on runoff forecasting system improvement, but there are several barriers that limit their implementation potential. The first barrier is the (still) limited availability of both meteorological and streamflow

data [18,67]. Although recent studies revealed pronounced biases in meteorological reanalysis products [68], as well as corresponding effects on the results of hydrological modeling [46,69], observation-based products are rarely readily available for use in operational forecasting systems. A similar situation is found with the availability of operational streamflow observations. Forecasting systems significantly benefit from the implementation of observational data assimilation routines [21,70]. However, the data often exhibit large errors and long discontinuities due to absent or insufficient quality control. The issue of the use of non-homogeneous meteorological data also remains unresolvable. Despite many research centers providing both reanalysis and forecast data, there is no openly available and spatiotemporal consistent dataset. The second barrier is more computational: there are many models that can be suitable in an operational flood forecasting framework at a continental scale [71], but the added complexity of their implementation related to the required data and computational resources may not be compensated by the obtained efficiency gains [72]. To some extent, this is also relevant to error correction techniques [73,74]. Thus, the development of runoff forecasting systems should continue to investigate the careful balance between the used data and models in terms of their availability, reliability, and computational complexity.

### 4.4. Website Traffic and Demand for Forecasts

Although timely and reliable runoff forecast production is the central focus of any forecasting service, the issued forecasts should be properly disseminated and communicated with the general public. OpenForecast v2 delivers runoff forecasts through the website (https://openforecast.github.io/, last access: 28 November 2020). There is an interactive map on the main page of this website that shows the location of all 834 gauges for which a runoff forecast is issued. Clicking on a gauge's icon on the map redirects the user to the individual webpage for the respective gauge with an interactive plot of the 7-day-ahead streamflow forecast (Figure 3).

Figure 11 demonstrates the time series of the daily number of unique visits to the OpenForecast main page. Results show that the first surge of interest in OpenForecast (16–25 March 2020) was related to its release, while the second surge was related to the flood period in Russia (April–June 2020). The dynamics of the website's traffic (Figure 11) underline the main difference between weather and runoff forecasts. Although the general public needs (and uses) weather forecasts on an everyday basis, runoff forecasts are needed only during flood periods. Thus, the demand for weather forecasts becomes a habit, while services for runoff forecasts remain marginal. This low and unsteady demand for runoff forecasts also limits their commercial potential, which is confirmed by the absence of private runoff forecast services. The low interest in runoff forecasts can be, to some extent, attributed to the weak communication skills of the hydrologists who develop the service. There are only a few papers (e.g., [50,75]) that put their central focus on communication rather than computational issues. Thus, hydrologists know how to compute forecasts but do not know how to communicate them efficiently.

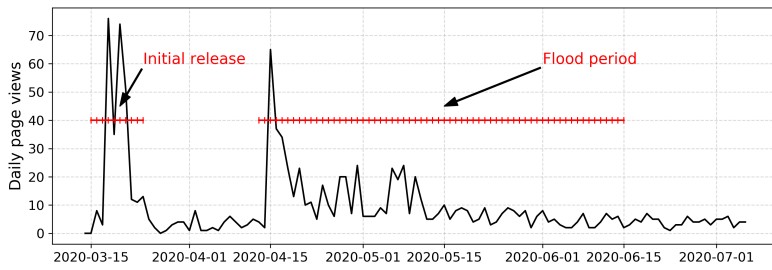

**Figure 11.** Daily attendance of the OpenForecast main webpage (https://openforecast.github.io/).

## 5. Conclusions and Outlook

In the presented study, the development and efficiency of OpenForecast v2—the first national-scale operational runoff forecasting system in Russia—are explored. OpenForecast v2 is fully based on open data and software and operationally delivers 7 day-ahead streamflow forecasts for 843 gauges across Russia. The verification of OpenForecast has been carried out using 244 gauges for which operational streamflow data were accessible and reliable during the verification period (14 March–6 July 2020). The skill of runoff forecasts, which measures the improvement in OpenForecast v2 over benchmark forecast, has been calculated and compared to two widely used benchmarks: climatology and persistence.

Results show that OpenForecast v2 provides reliable and skillful runoff forecasts up to one week. Thus, the loss of skill with lead time is more pronounced for small (<1000 km$^2$) river basins, rather than for medium (1000–10,000 km$^2$) and large (>10,000 km$^2$) basins. The results of the benchmark compared with climatology and persistence showed that OpenForecast v2 provides skillful predictions for most of the analyzed basins. Thus, it offers prominent additional value and has the potential to serve as a core component for early warning and water management systems.

The demonstrated development and benchmarking of OpenForecast v2 also helped to reveal several problems and questions worth exploring in future research. First, the results of the hydrological model calibration procedure showed that 161 out of 1004 basins (16%) did not pass the "skillful" thresholds in terms of NSE and KGE, and 118 out of the remaining 843 basins (14%) fell below the "satisfactory" threshold in terms of NSE. This shows that additional effort is needed both to understand the possible causes of model failure and to find respective ways to address the identified causes.

Second, an overall problem of data availability is shown. This problem mainly refers to Russian institutions, which are responsible for environmental monitoring. Thus, there is neither a model nor an observation-based, spatially and temporally consistent, national-scale meteorological dataset that can be used for hydrological applications. The situation with hydrological data is similar. Although the AIS database contains historical streamflow observations for more than 1000 gauges for the period of 10 years (2008–2017), it has not yet been updated with data from the preceding historical period. The ESIMO database, which maintains the circulation of the operational streamflow data, is designed in a way that limits its use in hydrological applications. Thus, the publication of streamflow data in Russia is considered rather exceptional than routine. A paradigm change is needed to improve the situation.

Third, there are obvious problems with runoff forecast communication. Many people in Russia may benefit from the timely and reliable forecasts produced by the OpenForecast system, but they do not know that it exists. The collaboration with media providers, such as TV channels, newspapers, web portals, and popular social networks, may help to improve the situation. However, additional (and substantial) resources are needed to arouse interest in cooperation.

In the first paper about the OpenForecast system [21], the authors mentioned that it has the potential to be implemented at a national scale. In the presented paper, this potential is exploited. Furthermore, the potential of the OpenForecast system is further elaborated upon, and it is argued that it could be further extended in the form of a framework: a set of open-source computational procedures that deploy operational runoff forecasting system for any region, country, or continent.

**Funding:** The reported study was funded by the Russian Foundation for Basic Research (RFBR) according to the research projects Nos. 19-05-00087 and 19-35-60005.

**Acknowledgments:** The author thanks Liubov Kurochkina for providing river basins' boundaries.

**Conflicts of Interest:** The author declares no conflict of interest.

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
