# Peer review of "OpenForecast v2: Development and Benchmarking of the First National-Scale Operational Runoff Forecasting System in Russia"

_hydrology, doi:10.3390/hydrology8010003_

Round 1

Reviewer 1 Report

     The authors are commended on the runoff forecasting system in Russia. However, runoff forecasting is often highly complex and heterogeneous. the topic of the manuscript relates to the scope of the journal. But I think I cannot recommend the manuscript to publish in the Journal. there are some parts should be improved.

The manuscript introduce development of the open forecast V2, but, there are many questiong in the manuscript, for example, how is developed about the open forecast?  also, how many paramertes in the model? how to calibrate the model?  and How to improve the precision about the model in the disscussion. 

Reviewer 2 Report

The findings are interesting and worth publishing. However, the author seems to be inexperienced in writing scientific papers and requires help with a rewrite. 

Reviewer 3 Report

The manuscript has many strong points. It’s a good application of hydrological science for operational forecasting and can be of great interest to global hydrological community. Additionally, It helps promote hydrological science through shared data and models. While the manuscript is generally organized well, it has many odd uses of English language. I strongly recommend editorial support to help improve English in the manuscript.

Reviewer 4 Report

General comments

In this manuscript the author presents an online interface developed to predict runoff at hundreds of stream gauges in Russia to help with runoff and flood forecasting. The results are provided free of charge for both water managers and the public. With the vast amount of data scientists are collecting globally, I believe the development of these types of modeling and forecasting tools will greatly benefit water managers in the future.  The author provided a good description of the framework and the data used for the development of the online interface and has put a considerable amount of effort into a modeling framework which can be easily improved in the future based on new available observations and data.

My main comment regarding this manuscript is related to the calibration section. The author should include more information on the models and how the modeling framework was employed? For example, what type of input data was used in these models besides weather data and watershed boundaries? Are the modeled watersheds split in hillslopes or sections, and if so, based on what type of approach? Are there soils considered and where is the data coming from? Is there a management or a land cover considered? Also, can these models be applied in a gridded mode? How is the weather data represented? Is the author using only one climate grid cell or multiple depending on the size of the basin? ERA5 is a very sparse data set (30 x 30 km2) while the basins are between 10 and 100 000 km2 so it would help to know how the climate is distributed in the modeled watersheds. Also, what were the calibrating parameters? Can the author provide a table with the ranges of the calibrated values? Also, do the calibrated values make sense? With 14 free parameters, we could match any system to some observed data. But do those parameters make physical sense? I do understand that this manuscript is manly presenting the modeling framework and the online interface, however, it would be helpful to have more information on the modeling approach and the calibration.  

There are too many sentences starting with “While”. Please reword some sentences to minimize the use of the word “While”. For example, see paragraph starting on line 362, though there are many more throughout the manuscript.

Where is equation 4 used? And what is r?

Section 4.1: The author should provide a list of all the calibrated parameters and add a small discussion regarding the calibration. Did all station require 14 or 6 calibrated parameters? Were there common parameters in most calibrated models?

Additional comments

Figure 3: Date is on x axis and Discharge is on y axis.

Figure 6: Should the legend read OpenForecast v2?

Figure 7: Is this figure for one or all the days in the verification period? Is it a snapshot calculated on a certain date (or seven days) for all stations or is it a snapshot calculated for the entire verification period?  For example, lead time = 1 is the average NSE calculated as an average of all daily predictions for the period of verification (n) and lead = 2 is the average NSE calculated as an average of all daily predictions for n+1 and so on?

Line 2: Does the author mean early flood warning?

Lines 5 – 7: Sentence not clear. Please reward.

Lines 13 - 14: The word “provide” has a positive connotation. Also, which sectors? Perhaps reward the sentence.

Line 16: Remove “in”.

Line 19-23: This is not a full sentence. Perhaps it would be better if it is merged with the previous sentence. Eg.:  “seasonal resolution [8–10]: the European Flood Awareness System…”.

Line 28: Replace “could yet be limited” to “is limited”.

Line 30: Remove “the potential for appropriate” as the sentence reads better without the extra words.

Line 40: Replace “none” with “no”.

Line 63: Remove the word “obtained”.

Lines 116 – 118. NSE and KGE tell us how well we are simulating the runoff peaks, however they don’t tell us if we are under- or over-estimating the total streamflow. We could have situations where the NSE is 0.50 but the %bias is off. Has the author considered also using %bias in the model calibration?

Line 196: Aperfect appears as a definition but it doesn’t appear in any equation.

Lines 222 – 224: How was this relationship developed if runoff data is available from 2008 to 2017 and the water level data is available for the last 7 days?

Lines 227 – 229: I understand the author’s reasoning for considering only the last 365 days for the discharge calculations, however, the rating curve will mainly change after large runoff events or floods. These events are easy to identify from the observed data and the author could chose to calculate the rating curves based on the years since the last major runoff or flood event.

Line 296: Replace “obtained with “the”.

Line 306: Delete “obtained”.

Line 316: Not sure “vivid” is the right word here.

Round 2

Reviewer 1 Report

 I think the authors cannot answer the questions clearly in the manuscript. 

Reviewer 2 Report

Only cosmetic changes have been made. The author requires outside help in presenting a publishable article.

Reviewer 4 Report

I thank the author for the additional information provided in the manuscript and for addressing all the comments.
